# Exploring stakeholder perceptions of peer support initiatives in the management of diabetes in low- and middle-income countries: An online survey study

Bishal Gyawali[1,2☯*], Soahum Bagchi[3☯], Zainab Dakhil[4], Israa Yaseen[5], Fathima Aaysha Cader[6], Lilian Pinto da Silva[7], Pooja Dewan[8], Diana Sherifali[9], Sheila Klassen[10]

1 Section for Health Services Research, Department of Public Health, University of Copenhagen, Copenhagen, Denmark, 2 Research Unit of General Practice, Department of Public Health, University of Southern Denmark, Odense, Denmark, 3 Harvard Medical School, Boston, Massachusetts, United States of America, 4 Ibn Al-Bitar Cardiac Centre, Al-Kindy College of Medicine/University of Baghdad, Baghdad, Iraq, 5 Baghdad Heart Center, Baghdad Teaching Hospital, Medical City, Baghdad, Iraq, 6 Kettering General Hospital, Kettering, United Kingdom, 7 Faculty of Physical Therapy, Federal University of Juiz de Fora, Juiz de Fora, Minas Gerais, Brazil, 8 British Heart Foundation Cardiovascular Research Centre, University of Glasgow, Scotland, United Kingdom, 9 School of Nursing, Faculty of Health Sciences, McMaster University, Hamilton, Ontario, Canada, 10 Division of Cardiovascular Medicine, Brigham and Women's Hospital, Boston, Massachusetts, United States of America

☯ Primary co-authors for this work
* bishalgyawali01@gmail.com

## Abstract

Peer support is an effective strategy to promote self-management behaviors and improve well-being in those with cardiometabolic disease, including type 2 diabetes mellitus (T2DM). There is limited knowledge about stakeholder perceptions regarding peer support programs in low- and middle-income countries (LMICs). The study assessed stakeholders' awareness and understanding of peer support initiatives for T2DM, and explored their perceived barriers and readiness for implementation. A cross-sectional, self-administered online survey with branching logic was distributed to stakeholders across macro- (health policy), meso (tertiary hospital), and micro (community) levels of LMIC healthcare systems from June 1 to December 15, 2023. Quantitative data were analyzed descriptively; qualitative data underwent thematic content analysis. A total of 69 respondents from 25 LMICs participated in the survey. Due to branching logic and response attrition, 53 surveys (77%) had complete responses. Most respondents were medical doctors (n = 35, 50.7%) and a large proportion worked in tertiary hospitals (n = 27, 39.1%). Thirty-nine respondents (56.5%) were aware of peer support; among the 38 respondents with complete data, 29 (76%) reported active involvement in T2DM peer support initiatives. Of 15 responses to open-ended questionnaires regarding barriers to T2DM peer support, 9 (60%) cited concerns about limited resources and lack of funding. Local leadership

**Data availability statement:** All relevant data are within the paper and its Supporting Information files.

**Funding:** This work was supported by the World Heart Federation (Grant Number WHF 2020/2021 awarded to the SPIDER research team). The funders had no role in study design, data collection and analysis, decision to publish, or preparation of the manuscript. No authors received salary support from the funder.

**Competing interests:** The authors have declared that no competing interests exist.

(mean ± standard deviation: 3.4 ± 1.2), resource allocation (2.7 ± 1.4), and sustainability planning (2.7 ± 1.4) showed the highest perceived readiness on a 5-point Likert scale (1 = strongly disagree, 5 = strongly agree). Stakeholders in LMICs demonstrate awareness and active involvement in T2DM peer support programs. While limited resources and funding remain significant barriers, local leadership, resource allocation, and sustainability planning showed the highest perceived readiness, indicating promising foundations for implementation. Strengthening these areas through targeted support could facilitate the expansion and sustainability of peer support initiatives in resource-constrained settings.

## Introduction

Type 2 diabetes mellitus (T2DM) is a significant global health concern due to its high prevalence and severe complications [1]. Effective management of T2DM requires continuous glucose monitoring, medication adherence, and lifestyle modifications, which can be especially difficult for patients and caregivers in resource-limited settings [2]. In these contexts, limited access to consistent care and education further complicates self-management efforts.

Peer support has emerged as a promising approach to enhance self-management among people living with T2DM [3]. By enhancing shared experiences and social connectedness, peer support offers emotional, informational, and practical assistance [4]. Evidence shows that peer support programs can improve medication adherence, promote lifestyle changes, enhance blood glucose monitoring, and offer crucial emotional support, helping patients navigate complex healthcare systems [5,6].

While peer support has demonstrated success in high-income countries [7], its role in low- and middle-income countries (LMICs) remains under-explored. A previous scoping review by our team highlighted inconsistencies in how peer support is defined and implemented for T2DM management in LMICs, with variations in terminology and practice potentially affecting its impact at the community level [8]. Furthermore, methods for integrating peer support into existing healthcare systems and understanding long-term outcomes and sustainability in these settings are unclear [8]. Addressing these gaps is critical to optimizing peer support implementation in resource-constrained environments.

Successful implementation of peer support requires understanding the awareness, experiences, and perceived readiness of key stakeholders, including healthcare providers, policymakers, community leaders, and people living with T2DM. The World Heart Federation's (WHF) Roadmap for cardiovascular disease prevention among individuals with T2DM underscores the importance of patient-centered approaches, including peer support [9,10]. However, practical implementation in LMICs has not been sufficiently examined. To address this, the current study aimed to assess stakeholders' awareness of peer support programs for T2DM management and to identify perceived barriers and readiness for implementation in LMICs.

## Materials and methods

### Ethics statement

This study involved anonymous, voluntary participation in an online survey with no collection of identifiable information. Based on these characteristics, the Mass General Brigham Institutional Review Board determined that the project does not constitute human subjects research and waived the requirement for formal IRB approval (REDCap ID#: 3211).

### Study design

A cross-sectional, self-administered online survey with branching logic was distributed to stakeholders living in LMICs. We used the LMIC definition from the World Bank [11]. The survey targeted stakeholders across these countries' macro, meso, and micro levels of health systems. Macro level stakeholders included representatives from ministries of health, private sector organizations, national diabetes associations, and national cardiac societies. Meso level stakeholders included those from regional health organizations, hospitals, clinics, and health centers. Micro level stakeholders consisted of physicians, non-physician healthcare professionals (e.g., community healthcare workers, and patient groups, and individuals living with T2DM themselves) [12]. The study adhered to the Checklist for Reporting Results of Internet E-Survey (CHERRIES) [13], ensuring comprehensive reporting of survey methodology. Survey links were sent in batches on a rolling basis to individual and organizational emails and response to links was not tracked. We included a disclaimer at the beginning of the survey clearly stating that their participation was entirely voluntary and anonymous, and there was no intent to link responses to identities as part of the study. Consent was implied by online completion of the survey. Participants were also informed that they could withdraw at any time without needing to notify the study team. We clearly stated that all data collected would be fully anonymized.

### Survey questions and domains

In spite of a thorough literature search, no suitable and validated questionnaire specific to study objectives was found. Therefore, we developed a self-administered questionnaire in English, comprising questions organized into three main sections: i) stakeholder's awareness and understanding of peer support initiatives for T2DM, ii) perceived barrier to implementation, and iii) readiness perception towards implementation. Response options were developed based on the findings of our scoping review [8], expert consultations, and discussion amongst the study team. Survey questions incorporated single and multiple-choice formats, Likert scale options for responses, and free-text fields to capture quantitative and qualitative responses. Readiness perceptions toward peer support initiatives were evaluated using questions rated on a 5-point Likert scale (5-Strongly agree, 4- Somewhat agree, 3-Neither agree nor disagree, 2-Somewhat disagree, 1-Strongly disagree). This readiness checklist consisted of 8 questions, where a higher mean score relative to the weighted mean indicated greater readiness. The survey was guided by SELFIE (Sustainable intEgrated chronic care modeLs for multi-morbidity: delivery, FInancing, and performancE) [14], an innovative framework advocating for the integration of micro, meso, and macro level elements within health systems, adapted slightly from six WHO health system components: service delivery, leadership and governance, workforce, financing, technology and medical products and information and research [14].

### Survey administration

The survey was purposely built for this study and was administered using Qualtrics™, a widely recognised internet-based survey platform known for its user-friendly interface. To ensure internal validity and reduce measurement error, the survey underwent a rigorous development process. It was initially reviewed by our multidisciplinary study team, which included professionals from medicine, public health, pharmacy, nursing and physiotherapy, all with experience conducting research in LMICs. The instrument was then refined based on feedback from an external panel of three experts to enhance content

validity and question clarity. Finally, the survey was pilot tested with a small group of healthcare professionals and program managers who were not part of the final sample. This process confirmed the functionality of the branching logic, established an acceptable completion time (approximately 10 minutes), and further improved the instrument's clarity and precision. Access to the survey was limited to invited participants via our sampling framework described below. Participants received a recruitment script via email along with a survey link. The survey was opened online on June 1, 2023, and closed on December 15, 2023. Reminder emails were sent after one (8th June 2023) and two weeks (15th June 2023) from the original invitation. No incentives were provided for participation. To maintain anonymity, no measures were taken to track responses and prevent multiple entries.

## Sampling

Recruitment utilized convenience and snowball sampling approaches facilitated through existing networks of the WHF, foundations, and organizations focusing on T2DM in LMICs. Where possible, participants within ministries of health were sought. Participants were invited to participate in the survey and encouraged to forward the invitation those in their networks who met the macro, meso, and micro criteria. As this was an exploratory, descriptive study without a primary hypothesis to test, no formal *a priori* sample size calculation was conducted. Instead, the target sample size was determined pragmatically with the aim of achieving a diverse representation of stakeholders across LMICs.

## Data analysis

Participants who completed less than 80% of the survey were excluded from the analysis. Categorical variables were examined using frequencies and percentages, while continuous data were assessed using mean and standard deviation (SD). Quantitative data analysis was performed using STATA version 17, Python 3, and Microsoft Excel. For qualitative items, a combined inductive-deductive thematic content analysis was conducted in Excel. The process began with an inductive review of the open-ended responses to identify emergent concepts. These initial concepts were then systematically organized and interpreted using the integrated SELFIE framework, which provided a deductive structure for categorizing findings across macro, meso and micro levels. Two authors (BG and SB) independently coded all responses, followed by iterative discussions to resolve discrepancies and refine the coding scheme until consensus was achieved. Codes were subsequently grouped into themes, with descriptive definition developed for each category. Although the brevity of the responses limited deeper exploration, thematic saturation was reached for the core barriers and facilitators relevant to the study objectives. The number of unique themes and the frequency of codes within each theme were also reported.

## Results

69 respondents completed the survey. Due to branching logic and attrition, 53 (77%) surveys had complete responses. Responses were received from 25 different countries, with the highest representation from Kenya (n = 9, 13.0%), followed by Brazil (n = 7, 10.1%) and Albania, Botswana, Malawi, Nepal, and Uganda (n = 4, 5.8%) depicted in **Fig 1**. There were 11, 36, and 22 respondents from the low, low middle, and high middle-income countries, respectively. The number of respondents from the respective countries and the country's income level, classified based on the World Bank data [11], is tabulated in **S1 Appendix**.

Most respondents were medical doctors (physicians) (n = 35, 50.7%), as shown in **Fig 2**. Respondents who responded others for their current job are listed in **S2 Appendix**. The majority of respondents were from tertiary healthcare facilities (n = 27, 39.1%), which include central hospitals, universities, and specialized hospitals as shown in **Fig 3**. Respondents were categorized by healthcare facility type: tertiary (central hospitals, universities, specialized hospitals), secondary (regional referral, provincial, regional, and district hospitals), primary (primary healthcare centers, health posts,

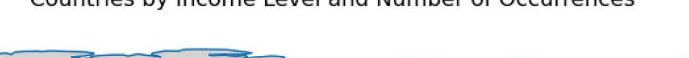

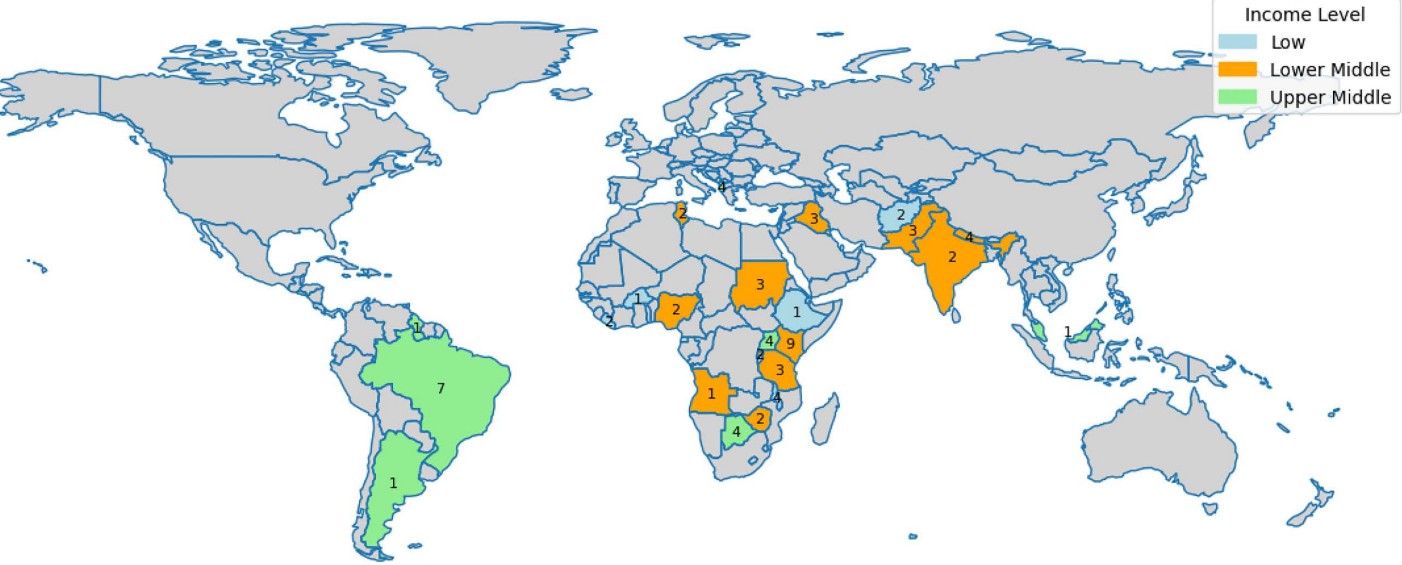

**Fig 1. Geographical distribution of survey respondents.** The map shows the distribution of respondents by country's income level (low-, lower-middle-, and upper-middle-income). A total of 69 respondents were recorded across these categories. In the low-income group (n = 11), respondents were from Afghanistan (2), Palau (1), Malawi (4), Ethiopia (1), Burkina Faso (1), and Liberia (2). The lower- middle-income group included 36 respondents from Nepal (4), Tunisia (2), Sudan (3), Rwanda (2), Pakistan (3), Nigeria (2), Angola (1), Zimbabwe (2), Kenya (9), Iraq (3), India (2), and Tanzania (3). The upper-middle-income group included 22 respondents from Malaysia (1), Brazil (7), Albania (4), Guyana (1), Argentina (1), Uganda (4), and Botswana (4). Geographic distribution of survey respondents across low- and middle-income countries. The map was generated using the open-source Python package GeoPandas (version 0.2.1; https://geopandas.org/). The underlying shapefile used by GeoPandas' built-in datasets is derived from Natural Earth (public domain; https://www.naturalearthdata.com/about/terms-of-use/).

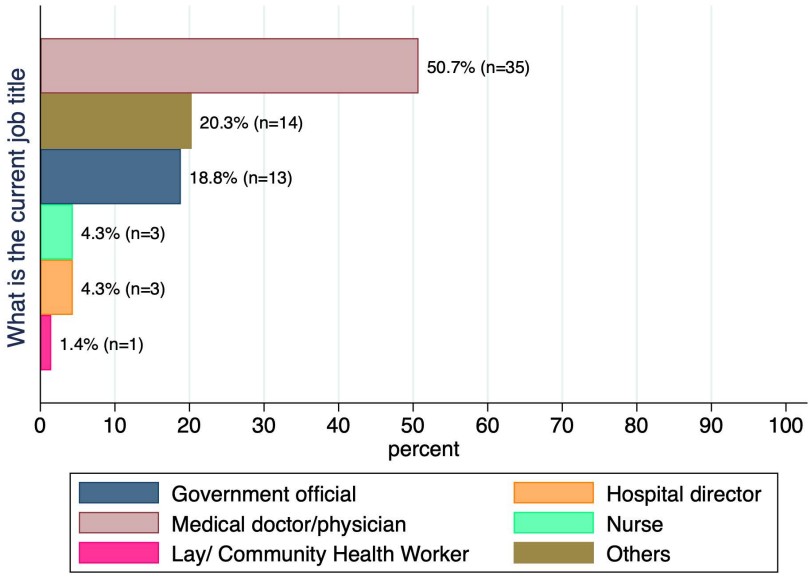

**Fig 2. Current job of the respondents.**

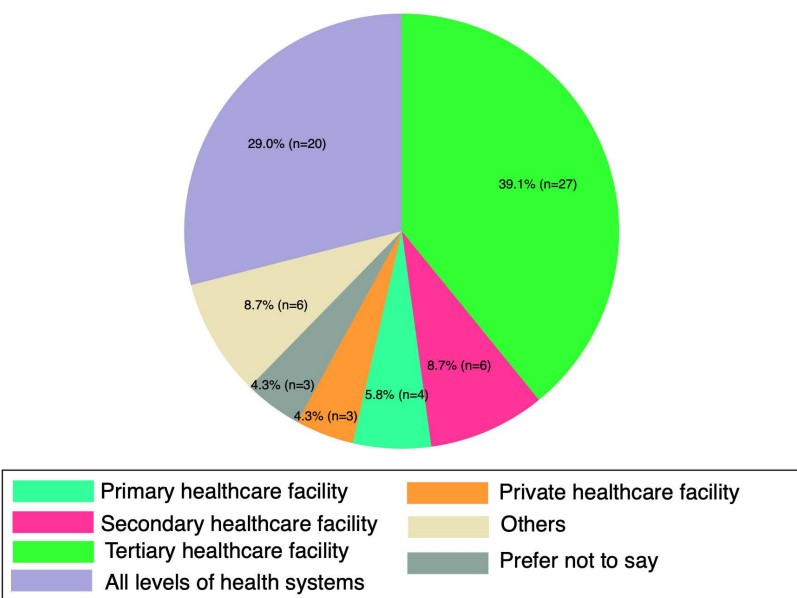

**Fig 3. Type of healthcare facility of respondents.**

dispensaries), and private (private hospitals, nursing homes, clinics, and faith-based institutions). Those who selected other for their facility type are listed in **S3 Appendix**.

### Awareness and understanding of peer support initiatives

The majority of respondents (n = 39/69, 56.5%) were aware of general peer support initiatives. Among these (n = 38, excluding one incomplete response), most (n = 23/38, 59%) were familiar with peer support initiatives aimed specifically at T2DM management. Of the 23 respondents familiar with peer support in T2DM, most were at the macro (n = 9; 39.1%) and micro levels (n = 9; 39.1%). A smaller proportion (n = 5; 21.8%) belonged to the meso level, as shown in **Fig 4**.

In response to the multiple-choice question, 'Who are the 'peers' in peer support initiatives in the management of T2DM?' (n = 18, 78.3%). Of the 23 respondents, it was indicated that peers were other individuals living with T2DM. Group peer support was most frequently utilized, with (n = 20, 86.9%) of 23 respondents indicating its use. Analysis of open-ended responses revealed respondents' varied interpretations of peer support, which included mutual support and understanding, providing emotional support and self-help, advocating for community rights, and sharing experiential knowledge depicted in **Table 1**.

### Barriers and facilitators to implementing peer support initiatives

Of 15 responses from open-ended questionnaires, multiple common barriers to peer support initiatives were identified: limited resources and lack of funding, poor program functioning, and lack of recognition within health facilities. Facilitators included financial support from various organizations, promotion and awareness, community resilience, and stakeholder collaboration. **Table 2** represents the full thematic content analysis, and **S5 Appendix** provides examples of the coding procedure used to develop **Tables 1** and **2**. The minimal data set supporting the results of this study is provided in **S6 Appendix**.

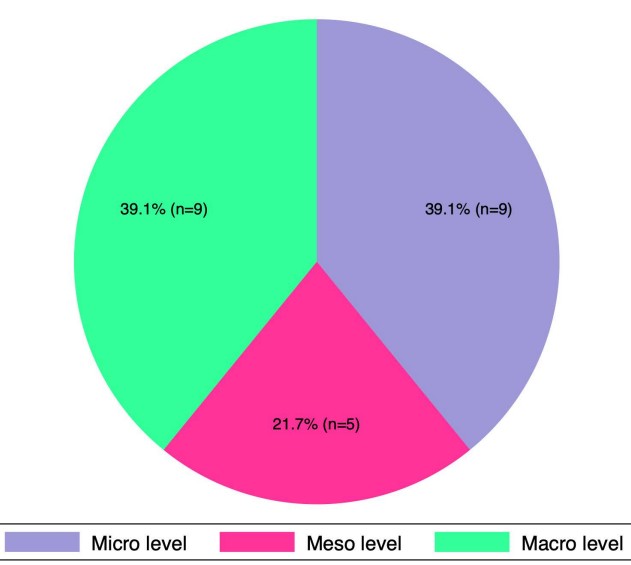

**Fig 4. Healthcare levels of peer support involvement.**

**Table 1. Thematic content analysis of stakeholder definitions of peer support.**

| Items | Themes (examples) | Frequency of themes |
|-------|-------------------|---------------------|
| How would you define peer support in your own words? | **Mutual support and understanding** (e.g.,: patient-to-patient support)<br>Description: Includes descriptions where individuals living with T2DM understand and assist each other through shared challenges and collaborative problem solving.<br>**Emotional support and self-help** (e.g.,: sharing personal experience for support)<br>Description: Includes definition that describes creating a supportive environment for individuals to express their feelings, share experiences, and develop coping strategies for emotional well-being.<br>**Community advocacy and rights** (e.g.,: social support network)<br>Description: Includes definition that involves advocating for the rights and needs of specific health conditions, which includes social support, ensuring equitable access to resources and addressing systemic issues.<br>**Experiential knowledge sharing** (e.g.,: shared experience support)<br>Description: Includes descriptions that focus on individuals offering support to others in similar situations by sharing their own experiences and insights. | 14<br>5<br>5<br>4 |

## Readiness perception of key stakeholders toward T2DM peer support initiatives

The readiness perception of key stakeholders toward implementing peer support initiatives for T2DM was generally favorable across three key domains: leadership willingness, resource allocation, and sustainability planning. Local leadership demonstrated strong perceived readiness, with 11 respondents (20.8%) strongly agreeing and 16 (30.2%) somewhat agreeing that they were willing to support the initiative, resulting in a mean score of 3.4 ± 1.2 on a 5-point Likert scale. Perceptions of readiness regarding the allocation of time and materials were also positive, with 8 respondents (15.1%) strongly agreeing and 7 (13.2%) somewhat agreeing, leading to a mean score of 2.7 ± 1.4. Similarly, 8 respondents (15.1%) strongly agreed and another 8 (15.1%) somewhat agreed on the adequacy of sustainability planning, yielding the same mean score of 2.7 ± 1.4. All three domains surpassed the predefined readiness threshold mean of 2.5, indicating an overall positive perception among stakeholders. These results are visually summarized in **Fig 5**, with corresponding confidence intervals shown, and additional statistical details available in **S4 Appendix**.

**Table 2. Thematic content analysis of stakeholder-identified barriers and facilitators to implementing peer support for T2DM in LMIC.**

| Items | Themes (examples) | Frequency of themes |
|---|---|---|
| Barriers | **Limited resources and funding** (e.g.,: organizations facing limited funding, equipment, facilities, trained personnel and inconsistent or insufficient funding) | 9 |
| | Description: Involves references to resource constraints and ongoing support for implementing peer support initiatives. | 4 |
| | **Poor program functioning** (e.g.,: lack of clear guidelines on program objectives, eligibility criteria of participation, types of supports offered, and referral processes) | 2 |
| | Description: Refers to the challenges faced in operating peer support initiatives effectively | |
| | **Lack of recognition within the healthcare facility** (e.g.,: lack of awareness among healthcare staff, insufficient resources, or staff time to support the implementation and maintenance of the peer support initiatives) | |
| | Description: Involves references to lack of recognition or support within the healthcare facility where peer support initiatives are implemented. | |
| Facilitators | **Financial support from various organizations** (e.g.,: continuous financial assistance from diverse organizations, such as the World Diabetes Foundation and UNICEF for ongoing support for diabetes peer support initiatives) | 5 |
| | Description: Involves references to financial support from various organizations for the growth and resilience of diabetes peer support initiatives, ensuring their long-term sustainability. | 4 |
| | **Promotion and awareness** (e.g.,: organizations launching a public awareness campaign to promote its peer support initiatives utilizing social media, organizing community events and conducting targeted outreach to reach potential participants) | 3 |
| | Description: Pertains to the efforts of organizations in raising awareness and advocating for implementing peer support initiatives to enhance care for diabetes communities. | 3 |
| | **Community resilience** (e.g.,: local organizations collaborating with community leaders and religious groups to disseminate important information about T2DM management options in community) | |
| | Description: Includes comments that refer to strong social bonds within the community and commitment to improving health outcomes, despite limited resources. | |
| | **Stakeholder collaboration** (e.g.,: medical professionals, government officials and non-government organizations coordinating workshops, offering resources, and fostering connections among individuals with T2DM) | |
| | Description: Includes comments that refer to the continuous involvement of various stakeholders' in supporting peer support initiatives within diabetes communities. | |

## Discussion

The findings of this study provide important insights into stakeholders' awareness, perceived barriers, and readiness for implementing peer support initiatives for T2DM in LMICs. Despite the study limitations listed below, we believe these preliminary and hypothesis-generating results are notable and warrant further study. The relatively high level of awareness and reported engagement indicate a promising foundation for peer support integration, suggesting that such programs are recognised at the grassroots level for their potential to enhance diabetes care. This is consistent with existing literature that has consistently shown that peer support can be valuable, cost-effective tool for improving glycemic control, self-care behaviors, , and overall well-being among people with T2DM, especially in resource-limited settings [3,5]. While these awareness levels are encouraging, they also emphasize the continued need for advocacy and education to expand the uptake and impact of peer support programs.

The preference for group-based peer support among participants further reinforces its practicality in LMIC contexts. Prior evidence indicates that group formats are particularly effective for enhancing shared experiences, emotional connection, and collective problem-solving, all of which are critical components in chronic disease management [15]. In

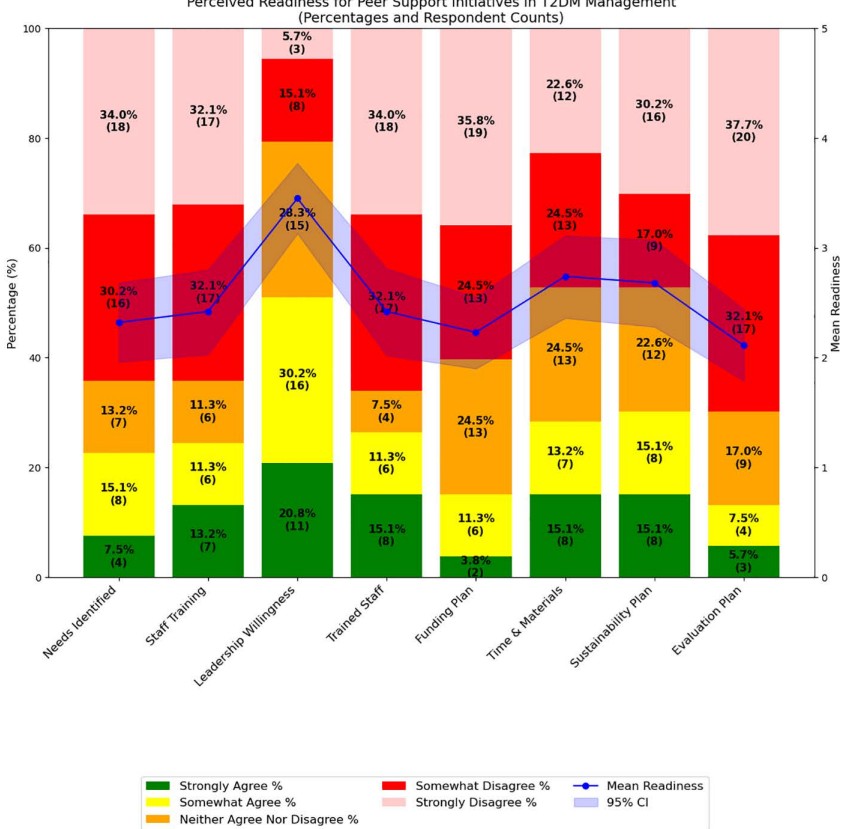

**Fig 5. Perceived readiness for T2DM peer support initiatives (5-point scale from 'strongly disagree' to 'strongly agree'), with mean readiness and 95% confidence interval (shaded blue).**

environments where stigma and systemic barriers persist, group support structures may offer a safe, empowering space for individuals to navigate their health journeys collectively. Additionally, group formats can efficiently disseminate information and create social cohesion, while one-on-one models may offer more personalized, private exchanges [16].

Stakeholders in this study commonly defined peer support as mutual support and understanding among individuals living with T2DM, involving shared challenges, and collaborative problem-solving. While our study includes a relatively limited sample of stakeholders, this still aligns with previous findings and our earlier scoping review, which emphasized the multifaceted nature of peer support, encompassing emotional support, advocacy, experiential knowledge sharing, and guidance on self-management behaviors like diet, exercise, and medication adherence [8,17]. These diverse roles make peer support adaptable and culturally responsive, which is particularly important in LMICs where standardized care models may not always be accessible or effective.

However, the most frequently cited barriers to implementation were limited resources and funding, including shortages of equipment, facilities, trained personnel, and inconsistent or insufficient funding. These findings echo earlier studies that identified inconsistent funding and logistical challenges as major threats to sustaining peer support programs [16,18]. Despite their low-cost reputation, successful and scalable peer support models still require foundational investments in training, coordination, and oversight [19,20]. In our study, most participants were affiliated with tertiary care centers, which may be more resource-rich and thus more suited to initiate such programs. However, for peer support to be equitable and far-reaching, implementation should eventually extend to rural and primary care settings.

In terms of readiness perception, stakeholders expressed generally favorable views across three key domains: local leadership support, resource allocation, and sustainability planning. Among these, leadership readiness appeared the most positively perceived, indicating a willingness to prioritize and support peer support initiatives at the organizational or institutional level. Perceived readiness regarding the availability of time and materials, and sustainability planning was also above the acceptable threshold, though somewhat less robust. These findings suggest a moderate to strong foundation for implementation, particularly when combined with targeted investments and planning support. The emphasis on leadership readiness aligns with broader research highlighting the critical role that local champions and institutional commitment play in sustaining community-based health interventions [21].

To our knowledge, this study is the first to comprehensively investigate stakeholders' awareness, perceived barriers, and readiness to implement peer support for T2DM across LMICs. The study was designed to target stakeholders across all LMICs as defined by the World Bank, enabling participation from a wide range of settings. The survey was designed using the SELFIE framework, enabling us to gather stakeholder perspectives at macro, meso, and micro levels. However, this study has several limitations. The modest sample size (n = 69) limits generalizability and raises the potential for non-response bias. Although purposive sampling enabled the inclusion of respondents from diverse countries, disciplines, and system levels, the small number within each subgroup means the findings should be considered preliminary and exploratory. The online survey format restricted participation to individuals with internet and computer access, potentially excluding those from less connected settings. This mode of recruitment may also have introduced selection bias by attracting participants with particular interest in the topic. Physicians were strongly represented in our sample (as physicians in LMICs are more likely to have internet and computer access), which may have skewed the findings toward the perspectives and priorities of medical doctors and underrepresented other stakeholders. A higher proportion of respondents were from tertiary care centers, which are typically more resourced and internationally connected than primary care or community-level providers, limiting the generalizability of findings to frontline and rural care contexts in LMICs. Although extensively piloted, the nature of our questionnaire may have introduced inconsistencies in how some respondents interpreted key concepts, particularly those less familiar with peer support. The survey was conducted in English, which may have limited participation from non-English-speaking stakeholders. Although patients were intended to be included as key stakeholders, none completed the survey. Given that patients are central to peer support initiatives, their absence limits the scope and applicability of the findings. This limitation also underscores the need for future studies to actively engage patients and peers in data collection to capture a more comprehensive understanding of stakeholder readiness. Finally, the qualitative data were derived from brief typed open-ended survey responses rather than in-depth interviews, constraining the richness and nuance of perspectives. Nonetheless, these responses yielded insights into salient barriers (e.g., lack of funding) and facilitators (e.g., local leaderships), which align with the exploratory aim of the study and provide direction for future research.

## Conclusions

Stakeholders in LMICs reported awareness and active involvement in peer support initiatives for T2DM. While limited resources and funding were commonly cited as key barriers to implementation, participants indicated favorable readiness across local leadership, resource allocation, and sustainability planning. These findings suggest a promising foundation for expanding peer support programs, particularly in tertiary care settings where such initiatives may currently be concentrated. Moving forward, efforts should focus on decentralizing these programs to rural and community settings. Future research should prioritize incorporating patient perspectives, alongside exploring effective delivery models, training strategies for peer supporters, and the integration of peer support into existing health systems. Strengthening these elements may enhance both the reach and sustainability of peer support initiatives in resource-limited settings.

## Supporting information

**S1 Appendix. Income level and number of respondents by country.**
(DOCX)

**S2 Appendix. Respondents with 'other' as job title.**
(DOCX)

**S3 Appendix. Respondents with 'other' as type of health care facility.**
(DOCX)

**S4 Appendix. Readiness perception of key stakeholders toward T2DM peer support initiatives.**
(DOCX)

**S5 Appendix. Example of coding process for thematic content analysis.**
(DOCX)

**S6 Appendix. Minimal dataset.**
(XLSX)

## Acknowledgments

We extend our deepest gratitude to all the participants who generously took the time to complete the online survey; their valuable insights were essential to this study. We also thank the WHF for providing the contact information of past Emerging Leaders (EL), which was instrumental in reaching our survey participants. The authors are grateful to the EL Faculty members, Prof. Amitava Banerjee and Prof. Pablo Perel, for their mentorship during the 2020/21 EL Program.

## Author contributions

**Conceptualization:** Bishal Gyawali, Zainab Dakhil, Fathima Aaysha Cader, Lilian Pinto da Silva, Pooja Dewan, Diana Sherifali, Sheila Klassen.

**Data curation:** Bishal Gyawali, Soahum Bagchi, Sheila Klassen.

**Formal analysis:** Bishal Gyawali, Soahum Bagchi.

**Funding acquisition:** Diana Sherifali.

**Investigation:** Bishal Gyawali, Soahum Bagchi, Zainab Dakhil, Israa Yaseen, Fathima Aaysha Cader, Lilian Pinto da Silva, Pooja Dewan, Diana Sherifali, Sheila Klassen.

**Methodology:** Bishal Gyawali, Soahum Bagchi, Zainab Dakhil, Israa Yaseen, Fathima Aaysha Cader, Lilian Pinto da Silva, Pooja Dewan, Diana Sherifali, Sheila Klassen.

**Project administration:** Bishal Gyawali, Diana Sherifali, Sheila Klassen.

**Resources:** Bishal Gyawali, Diana Sherifali, Sheila Klassen.

**Software:** Bishal Gyawali, Soahum Bagchi.

**Supervision:** Bishal Gyawali, Zainab Dakhil, Israa Yaseen, Fathima Aaysha Cader, Lilian Pinto da Silva, Pooja Dewan, Diana Sherifali, Sheila Klassen.

**Validation:** Bishal Gyawali, Soahum Bagchi, Sheila Klassen.

**Visualization:** Bishal Gyawali, Soahum Bagchi, Zainab Dakhil, Israa Yaseen, Fathima Aaysha Cader, Lilian Pinto da Silva, Pooja Dewan, Diana Sherifali, Sheila Klassen.

**Writing – original draft:** Bishal Gyawali.

**Writing – review & editing:** Bishal Gyawali, Soahum Bagchi, Zainab Dakhil, Israa Yaseen, Fathima Aaysha Cader, Lilian Pinto da Silva, Pooja Dewan, Diana Sherifali, Sheila Klassen.

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
