## [Decision Letter · Decision Letter 0]

16 Sep 2025

PGPH-D-25-01903

Exploring Stakeholder Perceptions of Peer Support Initiatives in the Management of Diabetes in Low- and Middle-Income Countries: An Online Survey Study

Dear Dr. Gyawali,

Thank you for submitting your manuscript to PLOS Global Public Health. After careful consideration, we feel that it has merit but does not fully meet PLOS Global Public Health’s publication criteria as it currently stands. Therefore, we invite you to submit a revised version of the manuscript that addresses the points raised during the review process.

Your manuscript has been evaluated by one reviewer, and their comments are available below. Please carefully revise your manuscript to address the concerns raised.

Please note that we have only been able to secure a single reviewer to assess your manuscript. We are issuing a decision on your manuscript at this point to prevent further delays in the evaluation of your manuscript. Please be aware that the editor who handles your revised manuscript might find it necessary to invite additional reviewers to assess this work once the revised manuscript is submitted. However, we will aim to proceed on the basis of this single review if possible.

We look forward to receiving your revised manuscript.

Kind regards,

Jenna Scaramanga

Staff Editor

Journal Requirements:

1. Please ensure that your Ethics Statement is available in its entirety at the beginning of your Methods section, under a subheading 'Ethics Statement'.

2. We note that your Data Availability Statement is currently as follows: “All relevant and available data is included in this paper.”

3. Some material included in your submission may be copyrighted. According to PLOS’s copyright policy, authors who use figures or other material (e.g., graphics, clipart, maps) from another author or copyright holder must demonstrate or obtain permission to publish this material under the Creative Commons Attribution 4.0 International (CC BY 4.0) License used by PLOS journals. Please closely review the details of PLOS’s copyright requirements here: PLOS Licenses and Copyright. If you need to request permissions from a copyright holder, you may use PLOS's Copyright Content Permission form.

Potential Copyright Issues:

Figure 1: please (a) provide a direct link to the base layer of the map (i.e., the country or region border shape) and ensure this is also included in the figure legend; and (b) provide a link to the terms of use / license information for the base layer image or shapefile. We cannot publish proprietary or copyrighted maps (e.g. Google Maps, Mapquest) and the terms of use for your map base layer must be compatible with our CC-BY 4.0 license.

Additional Editor Comments (if provided):

Reviewers' comments:

Reviewer's Responses to Questions

**Comments to the Author**

1. Does this manuscript meet PLOS Global Public Health’s publication criteria?

Reviewer #1: Partly

2. Has the statistical analysis been performed appropriately and rigorously?

Reviewer #1: No

3. Have the authors made all data underlying the findings in their manuscript fully available (please refer to the Data Availability Statement at the start of the manuscript PDF file)?

Reviewer #1: Yes

4. Is the manuscript presented in an intelligible fashion and written in standard English?

Reviewer #1: Yes

Reviewer #1: Strength of the study are,

• Peer support for T2DM management in LMICs is a relatively underexplored area,

• Application of the SELFIE framework provides a strong theoretical and conceptual base.

• Survey design, domains, sampling strategies, and administration are clearly reported, with transparency ensured through the CHERRIES checklist.

• Data collection spanning 25 LMICs enhances the breadth and generalizability of the study.

• Inclusion of varied healthcare settings—tertiary, secondary, primary, and private facilities—adds contextual depth and strengthens the findings.

Comments:

1. Sample Size Justification: The manuscript does not describe any attempt to calculate or justify the required sample size. Without a priori sample size estimation, it is unclear whether the study was adequately powered to capture the diversity of stakeholder perceptions across LMICs.

2. Internal Validity: Measures to ensure internal validity are insufficiently reported. Details regarding pilot testing of the tool and steps taken to reduce measurement error are lacking. These omissions reduce the methodological rigor.

3. Response Rate and Representativeness: The authors state that only 69 respondents completed the survey. This is a very low number considering the broad geographical scope across LMICs. Such a poor response rate introduces non-response bias and raises concerns about the representativeness and generalizability of the findings.

4. Thematic Content Analysis: Although the authors mention that thematic content analysis was used, the process of theme development is not clearly described. It is not specified whether themes emerged inductively from the data or were based on a pre-existing framework. Furthermore, no details are provided on coding procedures, inter-coder reliability, or whether data saturation was achieved. This lack of transparency undermines the credibility of the qualitative findings.

5. Sampling Bias: A majority of the respondents were physicians, which introduces sampling bias. Other important stakeholders—such as nurses, allied health workers, policymakers, and patients—were either underrepresented or excluded. This imbalance should be explicitly acknowledged in the limitations section.

6. Depth of Qualitative Data: The qualitative analysis appears to have been derived from brief survey responses rather than in-depth interviews. As a result, the richness and nuance of stakeholder perspectives are limited. This constitutes a methodological limitation that should be clearly highlighted.

7. Exclusion of Patients as Stakeholders: The study excludes patients, who are central to peer support initiatives. Without including their perspectives, it is difficult to conclude that “stakeholders in LMICs” as a whole demonstrate readiness for peer support initiatives. This limits the scope and applicability of the findings, and the conclusions should be tempered accordingly

**Do you want your identity to be public for this peer review?** For information about this choice, including consent withdrawal, please see our Privacy Policy

Reviewer #1: **Yes:** Dr.Kalaiselvan Ganapathy

---

## [Decision Letter · Decision Letter 1]

2 Jan 2026

Exploring Stakeholder Perceptions of Peer Support Initiatives in the Management of Diabetes in Low- and Middle-Income Countries: An Online Survey Study

PGPH-D-25-01903R1

Dear Dr. Gyawali,

We are pleased to inform you that your manuscript 'Exploring Stakeholder Perceptions of Peer Support Initiatives in the Management of Diabetes in Low- and Middle-Income Countries: An Online Survey Study' has been provisionally accepted for publication in PLOS Global Public Health.

Best regards,

Julia Robinson

Executive Editor

Reviewer Comments (if any, and for reference):

Reviewer's Responses to Questions

**Comments to the Author**

Reviewer #1: All comments have been addressed

Reviewer #2: All comments have been addressed

publication criteria?

Reviewer #1: Yes

Reviewer #2: Yes

3. Has the statistical analysis been performed appropriately and rigorously?

Reviewer #1: Yes

Reviewer #2: Yes

4. Have the authors made all data underlying the findings in their manuscript fully available (please refer to the Data Availability Statement at the start of the manuscript PDF file)?

Reviewer #1: Yes

Reviewer #2: Yes

5. Is the manuscript presented in an intelligible fashion and written in standard English?

Reviewer #1: Yes

Reviewer #2: Yes

Reviewer #1: All comments raised are addressed

Reviewer #2: This is a well written paper and flows very well.

My only concern is the contribution of authors from LMICs. Although the study is talking about LMICs, representation from LMICs is minimal and so is the contribution. This approach is being discouraged

**Do you want your identity to be public for this peer review?** For information about this choice, including consent withdrawal, please see our Privacy Policy

Reviewer #1: No

Reviewer #2: **Yes:** Patrick Dongosolo Kamalo
